# OpenReview forum: "Safety Anchor: Defending Harmful Fine-tuning via Geometric Bottlenecks"
_ICML.cc/2026/Conference — ICML 2026 regular_

### Official Review · Reviewer_jwib · 2026-02-22

**Soundness:** 3
**Presentation:** 3
**Significance:** 2
**Originality:** 2
**Overall Recommendation:** 4
**Confidence:** 3

**Summary:**

The authors work on designing better safety mechanisms to prevent harmful fine-tuning in such a way as to retain benefits from benign fine-tuning. To achieve this, they propose a defense that not only fine-tunes against the requested samples but also ensures that a fixed set of harmful prompts are anchored near to the refusal direction with an MSE loss. The authors perform a range of experiments across small-medium LLMs, and provide a range of ablation studies, and analysis. Their experiments clearly show that the proposed SBR performs much better on the harmfulness-helpfulness frontier.

**Compliance With Llm Reviewing Policy:**

Affirmed.

**Final Justification:**

Thanks to the authors for their efforts in preparing the response.

I am pleased to see that other reviewers had similar concerns, and the authors' new experiments, varying the poisoning ratio and analysis of computational overhead, address my main concerns.

I also appreciate the authors' new experiments on larger, more recent models in response to reviewer-CUkh, and i think this further supports the paper's claims.

Accordingly, I raise my score to a weak accept.

**Key Questions For Authors:**

* [q1] Why did the authors decide to use just 10% of the training data as harmful, and 90% harmless [L244,C1]? Why wouldn't an attacker use 100% harmful data? If the attacker is using purely harmful data, i suspect the anchor loss has a harder time being effective.
* [q2] what evidence or reasons do we have to believe the authors' claim on [L232,C1], that Llama3.1-8B etc are `"closest to the technology used in leading closed-source models"`? There are many possible valid reasons to use such a set of older models (cost, etc). However, I disagree with the authors' stated motivation, and would guess such models are likely not close to the leading closed-source models. If this is the motivation, perhaps we should be using the latest Kimi/Qwen models, or even the open source GPT-OSS reasoning models instead.

**Limitations:**

Limitations are briefly mentioned. A dedicated subsection should be written with an expanded discussion on these points raised, for any camera-ready version.

**Strengths And Weaknesses:**

# Strengths

## [s1] Clear problem formulation and solution

The authors do a good job at formulating the problem setting and threat models in the methodology. Specifically, the threat model in Section 3 is realistic and the authors' writing makes the setting clear and easy to understand. The proposed solution itself it simple and intuitive, which i regard as a clear strength of the paper.

## [s2] Wealth of ablation studies to explore the proposed method

The paper reports a number of ablation studies on the introduced hyper parameters and effect of various properties of the anchor prompts. Overall, I am fairly confident from these results that the method works relatively well under the specific formulation and specific split of harmfulness/harmless data (which i do have an important question about later on).


# Weaknesses

## [w1] There is an additional underexplored cost to anchoring the generations to the K anchor prompts
The authors' proposed loss fine-tuning objective in Eq. 4 includes an MSE loss against *all* the points in the safety set. I could not find an experiment thoroughly comparing reporting the computational cost as a function of the number of baseline prompts (or/and a theoretical analysis of FLOPs), compared to baseline methods. The authors discuss two numerical examples on [L322,c2], but i think this needs to be explored thoroughly. Practically speaking, end users (and potential adversaries) are going to choose a fine-tuning provider without defenses if the additional cost is significant.

## [w2] What if the attacker knows the anchor prompts, or uses 100% harmful data?

The authors partition the finetuning data and candidate safety anchor prompt pool into disjoint subsets. This raises an interesting question -- to what extent does the method still work if the attacker includes the safety anchor's prompts in the requested fine-tuning dataset (or even includes similar prompts)? In my understanding, the supervised finetuning objective may pull the safety anchors away from the desired refusal location. If this is true, an attacker can include many harmful prompts in their fine-tuning set that conflict with the objective of the anchor. Furthermore, relating to my question below, why doesn't an adversary simply include solely harmful examples?

## [w3] Organization of the first half of the paper makes it hard to read

I found the authors' writing structure to mean that the paper was unnecessarily hard to read.

Specifically, right after the introduction, the authors already begin to discuss a range of numerical results from their experiments around [L092, col2]. At this point in the paper, we do not yet have a clear sense of what the studied methods cited are doing, the precise technical formulation of their methodologies, and the adopted "Harmfulness Score" metric.

I would strongly suggest the authors consider beginning with the "Methodology" section after the introduction (as is the common convention). Here, the authors can formally introduce previous approaches in clear manner, before introducing their own methodology. Only after this, in the experimental section, should the authors discuss in detail such technical results. I believe it is perfectly fine to include a high-level, motivational discussion of the downsides of the related work in/after the introduction, but I feel many readers will be similarly confused when reading the paper in its current form, with such low-level discussion so early in the paper before the preliminaries are established.


## minor
* [L260,c1] "harmful" -> ``harmful''

---

> ### Author Rebuttal · Authors · 2026-03-27
>
> We thank the reviewer for acknowledging our method and ablation studies. We provide clarifications on evaluation protocols and additional stress tests below to address the reviewer's remaining concerns.
>
> ---
> ### **1: Underexplored computational overhead analysis.**
> We perform a computational overhead analysis across different anchor sizes, with comparisons against LISA and DeepAlign under A100. We define the training time of baseline HFT as 1x. The training time under SBR (1 Anchor) is 1.01x, indicating that it is 1.01 times that of HFT.
>
> **Table 1: Computational overhead comparison (Llama3.1-8B)**
> | Metric | HFT | SBR (1 Anchor) | SBR (8 Anchors, Default) | SBR (24 Anchors) | LISA | DeepAlign |
> |--------|--------------|-----------------|----------------------------|-------------------|------|-----------|
> | Peak GPU Memory (GB) | 16.25 | 16.54 | 16.58 | 18.89 | 17.12 | 25.34 |
> | Relative Training Time | 1× | 1.01× | 1.07× | 1.39× | 1.1× | 1.48× |
>
> With our default 8-anchor configuration, SBR introduces negligible overhead (only 7% longer training time, 0.33GB extra memory), which is lower than LISA and DeepAlign. This validates that SBR is lightweight for FaaS deployment.
>
> ---
> ### **W2 & q1: What if 100% harmful data**
> We adopted the 10%-30% poison ratio to align with established threat models in FaaS scenarios, consistent with prior defenses [1-3]. As evaluated in Section 4.2, this configuration aims to verify SBR's stability across varying practical attack intensities.
>
> To address the reviewer's question, we evaluate SBR under a 100% poison ratio, where poison examples n=100/500:
>
> **Table 2: Performance under 100% harmful data (Llama3.1-8B, Avg. HS (Harmful Score))**
> | Method | n=100 | n=500 |
> |---|---|---|
> | SFT | 74.2 | 78.6 |
> | DeepAlign | 17.7 | 59.5 |
> | Lisa | 71.5 | 76.7 |
> | SBR | 7.1 | 6.8 |
>
> As shown, SBR maintains an average HS < 9, while baselines collapse. This is consistent with our conclusion in Section 4.2 [L303, C1]: anchoring the geometric bottleneck effectively withstands aggressive malicious adaptation. The source code of SBR is available at: https://anonymous.4open.science/r/SBR-F876.
>
> ---
>
> ### **W2: What if Safety anchors in harmful fine-tuning dataset?**
> Following the disjoint setup in recent defenses [2, 4], we intentionally separate Safety Anchors from the training data to prevent data leakage and evaluate the zero-shot generalization of the defense.
>
> To address the reviewer's question, we test the scenarios where the attacker includes the anchors to override the defense. We sample anchors directly from the 300 harmful fine-tuning examples. We perform 3 independent runs for each anchor size.
>
> **Table 3: Performance when anchors are included in fine-tuning data**
> | Anchor Size | 1 | 8 | 24 |
> |-------------|---|---|----|
> | Avg. HS | 6.83(±0.32) | 4.63(±1.45) | 5.67(±1.16) |
> | Avg. FA | 93.96(±0.07) | 93.85(±0.06) | 93.75(±0.07) |
>
> As shown, SBR consistently maintains an average HS < 9 even under direct adversarial optimization. This result aligns with our conclusions in Section 4.3 [L315, C2]: SBR captures the universal semantics of refusal rather than overfitting to specific anchor patterns.
>
> ---
>
> ### **W3: Unconventional paper structure**
> We apologize for bringing a reading barrier. Our intention for Section 2 is to serve as an "Empirical Motivation". By exposing the failures of existing parameter-space defenses and investigating their underlying mechanisms, we naturally motivate the necessity of SBR. To resolve the readability issue, we will move Preliminaries to precede Motivation to ensure all concepts are clearly established upfront.
>
> ---
>
> ### **q2: Unsubstantiated claim**
> We select Llama3.1-8B as the primary evaluation model because it is widely used in LLM research, and it is more advanced than Llama2-7B, which is also widely used [1,2,4].
>
> We perform supplementary experiments on Qwen3-14B and Gemma3-13B. Due to space limitations, experimental results are detailed in our response to Reviewer CUkh (comment 2).
>
> We will delete the inappropriate sentence, replacing it with a rigorous explanation of our model selection rationale.
>
> ---
>
> We will also add a standalone Limitations section as suggested, rather than briefly mentioning it. We will correct the quotation mark typo at [L260, C1].
>
> We hope these discussions address your concerns, and we are pleased to provide further details if needed.
>
> ---
>
> [1] LISA: Lazy safety alignment for large language models against harmful fine-tuning attack. NeurIPS 2024
>
> [2] Vaccine: Perturbation-aware alignment for large language models against harmful finetuning attack. NeurIPS 2024
>
> [3] NLSR: Neuron-level safety realignment of large language models against harmful fine-tuning. AAAI 2025.
>
> [4] Booster: Tackling harmful fine-tuning for large language models via attenuating harmful perturbation. ICLR 2025.

---

> > ### Author Rebuttal · Reviewer_jwib · 2026-04-01
> >
> > Thanks to the authors for their efforts in preparing the response.
> >
> > I am pleased to see that other reviewers had similar concerns, and the authors' new experiments, varying the poisoning ratio and analysis of computational overhead, address my main concerns.
> >
> > I also appreciate the authors' new experiments on larger, more recent models in response to reviewer-CUkh, and i think this further supports the paper's claims.
> >
> > Accordingly, I raise my score to a weak accept.

---

> > > ### Author Response · Authors · 2026-04-02
> > >
> > > Thank you for reviewing our rebuttal and updating the score. We are glad to know your main concerns have been resolved.

---

### Official Review · Reviewer_oSNo · 2026-03-06

**Soundness:** 3
**Presentation:** 3
**Significance:** 3
**Originality:** 3
**Overall Recommendation:** 5
**Confidence:** 3

**Summary:**

This paper presents a defense against harmful fine-tuning of LLMs. First, the authors’ demonstrate that LLMs contain a dense number of harmful directions in their parameter and activation spaces, which limits the effectiveness of prior defenses that attempt to constrain weight or activation shifts. Motivated by this, the authors’ introduce Safety Bottleneck Regularization (SBR), which instead focuses on the unembedding matrix, a natural “geometric bottleneck”. Specifically, SBR anchors a model’s last-layer hidden representation on harmful queries to pre-computed representations of refusal, such that unembedding harmful representations still enforces aligned outputs. Experiments demonstrate that SBR offers significantly more resilience to harmful fine-tuning than prior approaches while maintaining task performance.

**Compliance With Llm Reviewing Policy:**

Affirmed.

**Final Justification:**

The proposed method achieves SoTA performance against harmful fine-tuning via a novel approach.

Through the rebuttal process, the authors addressed all of my outstanding concerns with strong empirical results, so I am inclined to raise my score from a Weak Accept to Accept.

**Key Questions For Authors:**

The number for each question corresponds to the weakness above.

1. Is SBR effective at higher poisoning rates (including 1.0)?

2. Is SBR effective if the attacker intentionally chooses harmful queries that the base model does not refuse to answer?

3. Is SBR effective if the safety anchors are chosen from a different distribution than the harmful fine-tuning dataset (e.g., HarmBench [1] vs. BeaverTails [2])?

4. Can the author’s extend their experimentation to a full-scale fine-tuning setting to verify that the achieved results are not due to limitations of LoRA fine-tuning?

[1] Mazeika et al., “HarmBench: A Standardized Evaluation Framework for Automated Red Teaming and Robust Refusal,” ICML 2024.

[2] Ji et al., “BeaverTails: Towards Improved Safety Alignment of LLM via a Human-Preference Dataset,” NeurIPS 2023.

**Limitations:**

Yes

**Strengths And Weaknesses:**

### **Strengths**

1. The paper conducts a thorough analysis of prior defense paradigms, demonstrating that the dense number of harmful directions in LLM’s parameter and activation spaces limits their effectiveness. This provides strong motivation for the proposed approach and also demonstrates its originality.

2. SBR is significantly more effective than the evaluated baselines, achieving $<10$ HS in all evaluated settings, even under backdoor-based adversaries.

3. SBR appears simple and easy to integrate into standard fine-tuning pipelines, making it a practical solution to mitigate harmful fine-tuning. Furthermore, it requires only a few ($\leq 8$) safety anchors, implying negligible overhead.

4. The paper is well written and easy to follow. The proposed method is clearly presented without any (noticeable) lack of important details.

### **Weaknesses**

1. Per the assumed problem scenario, the attacker has full control over their (private) training dataset. If their goal is to unalign the base model, it seems they would likely just use a dataset consisting entirely of harmful queries. However, the paper evaluates only up to a harmful proportion of 0.3, making it unclear whether SBR is effective against this (practical) attack.

2. As the paper mentions, SBR relies on the initial model having a strong base level of alignment. One concern I have is that if the attacker can find harmful training examples that the base model doesn’t reject, these may bypass SBR and compromise the fine-tuned model’s overall safety. If this is the case, SBR may be defeated via a simple pre-processing step from the attacker.

3. SBR’s effectiveness depends on the safety anchors generalizing to harmful instructions in the attacker’s training data. The experimental setup guarantees this by sampling all harmful data from the same benchmark. However, in practice, the defender does not have access to the training data when selecting safety anchors. It is unclear whether SBR would remain effective in the presence of distribution shifts in harmful data sources.

4. The evaluation uses LoRA fine-tuning, effectively constraining the rank of parameter updates to $r=16$. This likely instantiates a weaker attacker than standard full-scale fine-tuning.

---

> ### Author Rebuttal · Authors · 2026-03-27
>
> We thank the reviewer for recognizing the value of our work and for the constructive feedback. We hope the following clarifications and additional stress-test experiments can fully address your concerns.
>
> ---
>
> ### **Comment 1: Add experiments with 100% poison data ratio**
> We adopted the 10%-30% poison ratio to align with established threat models in FaaS scenarios, consistent with prior defenses [1-3]. As evaluated in Section 4.2, this configuration aims to verify SBR's stability across varying practical attack intensities.
>
> To address the reviewer's question, we evaluate SBR under a 100% poison ratio, where poison examples n=100/500:
>
> **Table 1: Performance under 100% harmful data (Llama3.1-8B, Avg. HS (Harmful Score))**
> | Method | n=100 | n=500 |
> |---|---|---|
> | SFT | 74.2 | 78.6 |
> | DeepAlign | 17.7 | 59.5 |
> | Lisa | 71.5 | 76.7 |
> | SBR | 7.1 | 6.8 |
>
> As shown, SBR maintains an average HS < 9, while baselines collapse. This is consistent with our conclusion in Section 4.2 [L296, C1]: anchoring the geometric bottleneck effectively withstands aggressive attack.
>
>
> ---
>
>
> ### **Comment 2: Harmful training examples that the initial model does not reject**
> We thank the reviewer for this interesting comment. We clarify that SBR is designed to **preserve the existing safety guardrails of pre-aligned models**, not to endow safety capabilities to models with no initial alignment. This is consistent with our focused FaaS scenario, where the base models hosted by service providers are all well safety-aligned. Thus, attackers struggle to collect sufficient harmful samples that bypass initial refusals for effective malicious fine-tuning.
>
>
> ---
>
> ### **Comment 3: Distribution shift between safety anchors and harmful data**
> Because mainstream harmful datasets share a high similarity in the semantic distribution of malicious intents, we initially follow recent FaaS defense protocols [1-3] by utilizing BeaverTails.
>
> To address the reviewer's concern, we conduct cross-dataset evaluations across BeaverTails, AdvBench, and HarmBench (each repeated three times, and we report the mean and standard deviation).
>
> **Table 2: Cross-distribution generalization (poison ratio p=0.3, Llama3.1-8B)**
> | Setting | Anchor Source | Train Data | Test Data | Avg. HS | Avg. FA |
> |---------|---------------|------------|-----------|---------|---------|
> | HFT Baseline | - | - | - | 74.3 | 94.38 |
> | SBR | BeaverTails | BeaverTails | BeaverTails | 8.2 | 94.03 |
> | SBR | BeaverTails | AdvBench | HarmBench | 10.33(±1.02) | 93.46(±0.12) |
> | SBR | HarmBench | BeaverTails | AdvBench | 9.67(±0.85) | 93.73(±0.07) |
> | SBR | AdvBench | HarmBench | BeaverTails | 8.5(±1.32) | 94.00(±0.07) |
>
> SBR maintains HS<11 even when anchors and attack data come from different datasets. This supplementary test yields no contradictory findings; rather, it reinforces our geometric analysis in Section 4.3 [L315, C2]: SBR captures the universal semantics of refusal rather than overfitting to specific anchor patterns. Consequently, anchoring this specific geometric bottleneck allows SBR to generalize robustly, regardless of the specific dataset used.
>
> ---
>
> ### **Comment 4: Evaluate full fine-tuning instead of only LoRA**
> We utilize LoRA to ensure a fair comparison, as it is the standard evaluation protocol established by our baselines [1, 2, 3]. Our primary evaluations under this setting adequately demonstrate how attackers exploit parameter redundancy.
>
> To address the reviewer's question, we perform full fine-tuning experiments on Llama3.1-8B, where lr=1e-5, λ=50, covering n=100/300 pure poison examples:
>
> **Table 3: Performance under full fine-tuning (Llama3.1-8B)**
> | Method | HS$_{n=100}$ | HS$_{n=300}$ |
> |--------|---------------|---------------|
> | HFT (no defense)   | 76.6          | 78.4          |
> | SBR    | 14.7          | 23.8          |
>
> Even under a full fine-tuning attack, SBR still maintains a significant defensive advantage. This is consistent with our conclusion in Section 4.2 [L253, C2]: SBR’s geometric bottleneck mechanism effectively inhibits the generation of harmful content.
>
> ---
>
> We hope these address all the concerns. We are pleased to provide further details if needed.
>
> ---
>
> [1] LISA: Lazy safety alignment for LLMs against harmful fine-tuning attack. NeurIPS 2024
>
> [2] Vaccine: Perturbation-aware alignment for LLMs against harmful finetuning attack. NeurIPS 2024
>
> [3] NLSR: Neuron-level safety realignment of LLMs against harmful fine-tuning. AAAI 2025.

---

> > ### Author Rebuttal · Reviewer_oSNo · 2026-04-02
> >
> > I thank the authors for their responses and find that most of my concerns have been resolved.
> >
> > However, I think comment 2 has not been fully addressed. I do not assume that the initial model is non-aligned; rather, that there is a small set of harmful examples that it complies with due to limitations in alignment training. For instance, [1] shows that, even without an attack, Llama3-8B still answers ~10% of harmful queries in their setup. In this case, the attacker could easily identify such a subset (e.g., from BeaverTails) and then train on it, rather than on randomly sampled harmful instances. I wonder if the authors could test whether safety can be preserved in this setting?
> >
> > If this can be addressed, I would be happy to raise my score.
> >
> > ---
> >
> > [1] Yu et al., "Robust LLM safeguarding via refusal feature adversarial training," ICLR 2025.

---

> > > ### Author Response · Authors · 2026-04-03
> > >
> > > We thank the reviewer for the clarification and the referenced literature.
> > >
> > > To address the reviewer's concern, we conduct the following evaluation on Llama3.1-8B:
> > > 1. We evaluate the original model on the BeaverTails training set and identify harmful queries it inherently fails to refuse.
> > > 2. We randomly sample 100 such "non-refused" queries to replace the random malicious samples in our original training mixture, forming a 1,000-sample dataset (poison rate $p=0.1$). Meanwhile, we ensure these "non-refused" queries are disjoint from our test set to prevent data breaches.
> > > 3. We evaluate SBR and SFT under this targeted setup using our standard test set.
> > >
> > > **Table: Performance under Targeted Attack using Native Vulnerabilities**
> > >
> > > | Setting | Harmful Score (HS) |
> > > | :--- | :--- |
> > > | Base Model (No HFT) | 3.4 |
> > > | SFT (Targeted Attack) | 59.3 |
> > > | SBR (Targeted Attack) | 4.1 |
> > >
> > > As shown, when an attacker exploits the model's native alignment flaws, standard SFT suffers a safety collapse (HS surges to 59.3). In contrast, SBR maintains a low HS＝4.1. This demonstrates that SBR anchors existing alignment, preventing targeted attacks from tearing open initial vulnerabilities.
> > >
> > > We hope this resolves the reviewer's remaining concern.

---

### Official Review · Reviewer_8kA6 · 2026-03-09

**Soundness:** 3
**Presentation:** 3
**Significance:** 3
**Originality:** 3
**Overall Recommendation:** 5
**Confidence:** 3

**Summary:**

In this manuscript, the authors investigate the issue of harmful fine-tuning and observe that existing parameter-space defenses fail due to the high-dimensional redundancy of model parameters. Based on this insight, The author proposes shifting the focus of defense to the model’s output layer. Specifically, they introduce a regularizer called SBR, which constrains the deviation of the final hidden state in the last layer on harmful samples from its originally aligned representation during fine-tuning, thereby achieving defensive effects. The authors further validate the effectiveness of SBR through extensive experiments, including evaluations on four downstream datasets and three model architectures. The topic is of considerable significance for advancing large language model training and safety alignment. However, several aspects require further clarification and improvement to enhance the overall rigor and completeness of the paper.

**Compliance With Llm Reviewing Policy:**

Affirmed.

**Final Justification:**

The authors have conducted additional experiments on AdvBench and also evaluated their method on the updated Gemma3-13B model, which addresses all of my concerns. It is also encouraging to see that the authors have resolved several key issues raised by other reviewers through further experiments.

I vote to accept this work.

**Key Questions For Authors:**

- The authors only evaluate their method on a single harmful dataset BeaverTails. It is suggested to include additional harmful datasets, such as AdvBench and HarmBench, to provide a more comprehensive evaluation.
- The experiments lack analysis of computational overhead, such as runtime and peak GPU memory consumption for different methods.
- It is suggested that the authors include a framework diagram in the methodology section to more intuitively illustrate the overall approach.
- The related work section is missing discussion of several recent studies, such as Pharmacist, LoX, AntiDote, SPARD, SC-LoRA, and Surgery. The authors may refer to https://github.com/git-disl/awesome_LLM-harmful-fine-tuning-papers to further organize and improve this section.
- The Gemma-1.1-7B model used in the paper is relatively outdated. It is suggested to conduct experiments with a more recent model, such as Gemma-3-12B, to strengthen the empirical validation.
- The paper should include a discussion on the limitations of the proposed approach.

**Strengths And Weaknesses:**

**Strengths:**
- In the manuscript, the authors analyze the limitations of existing defense methods and propose a regularizer based on the final layer to mitigate harmful fine-tuning. The experimental results demonstrate strong defensive performance.
- The paper is clearly written and includes a relatively comprehensive theoretical analysis and experimental validation to support the effectiveness of the proposed method.

 **Weaknesses:**
- Please refer to the Key Questions for the Authors listed above.

---

> ### Author Rebuttal · Authors · 2026-03-27
>
> We thank the reviewer for acknowledging our analysis, writing, and experimental validation. We also thank the reviewer's constructive feedback. We hope the following discussions can resolve the reviewer's remaining concerns.
>
> ---
>
> ### **Comment 1: Add AdvBench and HarmBench for comprehensive evaluation**
> We utilize BeaverTails as our primary benchmark to align with the evaluation protocols established by recent mainstream defenses [1-3]. Because BeaverTails provides a comprehensive distribution of malicious intents in the community, evaluating on it is sufficient to support our core insight.
>
> To address the reviewer's suggestion, we evaluate SBR on AdvBench:
>
> **Table 1: AdvBench full task evaluation (p=0.1, Llama3.1-8B, HS/FA)**
> | Method | SST-2 | AGNEWS | GSM8K | AlpacaEval |
> |--------|-------|--------|--------|------------|
> | HFT    | 74/94.15 | 78/91.1 | 77/82.6 | 74/44.39 |
> | Lisa   | 58/94.27 | 53/90.3 | 44/76.8 | 55/39.78 |
> | DeepAlign | 21/92.89 | 27/89.7 | 24/81.7 | 31/37.61 |
> | SBR (Ours) | 6/93.81 | 5/90.2 | 4/81.0 | 8/42.76 |
>
> As shown, SBR consistently maintains a low Harmful Score (HS ≤ 8) while preserving high Functional Accuracy (FA) across all downstream tasks on AdvBench. This supplementary evaluation yields results that are consistent with our primary findings on BeaverTails: anchoring the geometric bottleneck is a universally effective defense mechanism.
>
> ---
>
> ### **Comment 2: Computational overhead analysis**
> We perform a computational overhead analysis across different anchor sizes. We define the training time of baseline HFT as 1x. The training time under SBR (1 Anchor) is 1.01x, indicating that it is 1.01 times that of HFT.
>
> **Table 2: Computational overhead comparison (Llama3.1-8B, single A100)**
> | Metric | Baseline HFT | SBR (1 Anchor) | SBR (8 Anchors) | SBR (24 Anchors) | LISA | DeepAlign |
> |--------|--------------|-----------------|------------------|-------------------|------|-----------|
> | Peak GPU Memory (GB) | 16.25 | 16.54 | 16.58 | 18.89 | 17.12 | 25.34 |
> | Relative Training Time | 1× | 1.01× | 1.07× | 1.39× | 1.1× | 1.48× |
>
> Our computational cost is primarily influenced by the number of safety anchors. Under our default setting of 8 anchors, the additional computational cost introduced is lower than that of both baselines.
>
> ---
>
> ### **Comment 3: Lack of framework diagram in Methodology**
> The core insight of applying a constraint is illustrated in Figure 1. However, we agree that adding a framework diagram makes the methodology more intuitive. We design a schematic diagram outlining SBR's workflow. It can be viewed at: https://anonymous.4open.science/r/overview_image/README.md
>
> ---
>
> ### **Comment 4: Adding related work**
> We have carefully read all the mentioned recent work and will supplement the Related Work. Pharmacist & SPARD are data-centric defense perspectives; LoX & AntiDote are post-hoc remediation paradigms; Surgery, SC-LoRA are parameter-space robustness optimization. In contrast, SBR shifts the defense focus to the deterministic unembedding geometric bottleneck, which avoids the orthogonal attack bypass problem.
>
> ---
>
> ### **Comment 5: Outdated Gemma-1.1-7B, suggest newer Gemma3**
> Our initial choice of Gemma-1.1-7B followed the SOTA baseline, DeepAlign[4], to ensure a fair and direct comparison. While Gemma-1.1-7B is earlier, the fundamental unembedding geometry remains consistent across Transformer-based LLMs; thus, it does not affect the validity of our core insight.
>
> Following the reviewer's suggestion, we evaluate SBR on the newer Gemma3-13B:
>
> **Table 3: Gemma3-13B defense performance**
> | Method | HS$_{p=0.1}$ | HS$_{p=0.3}$ | Avg. FA |
> |--------|---------------|---------------|---------|
> | SFT    | 77.8          | 80.4          | 94.10   |
> | DeepAlign | 19.5        | 35.2          | 91.80   |
> | LISA   | 58.4          | 71.3          | 93.47   |
> | SBR (Ours) | 8.0         | 12.3          | 93.54   |
>
> As shown, SBR maintains HS < 13 across attack intensities, while baselines still struggle. These results are consistent with our existing findings on [L309, C1]: SBR does not rely on specific model versions and generalizes to other Transformer-based LLMs.
>
>
> ---
>
> ### **Comment 6: Limitations**
>
> We briefly discussed the limitations in the Conclusion. Following the reviewer's suggestion, we will create a standalone "Limitations" section from the following perspectives:
> (1) SBR relies on the initial safety alignment of the LLM; (2) SBR does not defend against inference-time jailbreaks (e.g., GCG) because it does not alter model parameters.
>
> ---
>
> We are pleased to provide further details if needed.
>
> ---
>
> [1] LISA: Lazy safety alignment for LLMs against HFT attack. NeurIPS 2024
>
> [2] Vaccine: Perturbation-aware alignment for LLMs against HFT attack. NeurIPS 2024
>
> [3] NLSR: Neuron-level safety realignment of LLMs against HFT. AAAI 2025.
>
> [4] Safety alignment should be made more than just a few tokens deep. ICLR 2025

---

> > ### Author Rebuttal · Reviewer_8kA6 · 2026-04-02
> >
> > Thanks to the authors for their detailed response.
> >
> > The authors have conducted additional experiments on AdvBench and also evaluated their method on the updated Gemma3-13B model, which addresses all of my concerns. It is also encouraging to see that the authors have resolved several key issues raised by other reviewers through further experiments.
> >
> > Accordingly, I raise my score to Accept.

---

> > > ### Author Response · Authors · 2026-04-02
> > >
> > > We appreciate your recognition of our work. We are glad our response has resolved your concerns and thank you for increasing the score.

---

### Official Review · Reviewer_CUkh · 2026-03-13

**Soundness:** 4
**Presentation:** 4
**Significance:** 3
**Originality:** 2
**Overall Recommendation:** 4
**Confidence:** 4

**Summary:**

This paper proposes a simple defense technique against harmful fine-tuning attacks called Safety Bottleneck Regularization (SBR) in Fine-Tuning as a Service scenarios. SBR employs a simple MSE regularization loss term to keep the final layer hidden represenations for harmful inputs during fine-tuning close to that of the original safety-aligned model. Experimental results on Llama 3.1 8B, Qwen 2.5 7B and Gemma 1.1 7B show that harmfulness can be effectively reduced without sacrificing model utility. These findings remain consistent across various poisoning ratios and more stealthy backdoor attacks.

**Compliance With Llm Reviewing Policy:**

Affirmed.

**Final Justification:**

Final recommendation: 4 (weak reject). My initial concerns were full addressed, but I’ve decided to keep my original rating.

**Key Questions For Authors:**

How effective would SBR be against input-space jailbreak attacks such as GCG [1]? It could be interesting to see if there are any extended benefits in that direction beyond fine-tuning attacks.

[1] Zou, Andy, et al. "Universal and transferable adversarial attacks on aligned language models." arXiv preprint arXiv:2307.15043 (2023).

**Limitations:**

yes

**Strengths And Weaknesses:**

Strengths:
1. The regularization method is simple, computationally-efficient and can be easily integrated into existing fine-tuning frameworks.
2. The ablation studies are quite extensive, and provide an interesting thorough analysis on the impact of SBR.
3. Theoretical analysis is provided to support the argument about why protecting models through regularization on the parameter space is insufficient, helping to motivate their unembedding-based regularization.

Weaknesses:
1. As shown in [1], the evaluation results for defense effectiveness can vary widely depending on experiment configuration, such as the learning rate used during fine-tuning. It would be good to perform some ablations on this aspect to see if SBR remains robust.
2. The model selection is OK, Llama 3.1 is fairly recent but it would be good to see results for newer versions of Qwen (e.g., 3 or 3.5) and Gemma (e.g., 3). Also, it would be good to explore both larger (> 8B) and smaller (< 7B) models to see if SBR is still effective in more extreme model sizes.

---

> ### Author Rebuttal · Authors · 2026-03-27
>
> We thank the reviewer for acknowledging our analysis, method, and ablation studies. We appreciate the reviewer's constructive feedback and have addressed all comments. We hope the following clarifications and supplementary results can resolve the reviewer's concerns.
>
> ---
>
> ### **Comment 1: Ablation on fine-tuning learning rate**
> We utilize a learning rate of 1e-5 to ensure a fair and direct comparison, following the mainstream configuration established by baseline defenses [1, 2].
>
> To address the reviewer's concern and further explore the hyperparameter robustness boundaries, we conduct an ablation on learning rates $5\times10^{-5}$ and $10^{-4}$ under poison rate $p=0.1$:
>
> **Table 1: All methods across LR (Llama3.1-8B, $p=0.1$)**
> | Method       | HS$_{1e-5}$ | HS$_{5e-5}$ | HS$_{1e-4}$ | FA$_{1e-5}$ | FA$_{5e-5}$ | FA$_{1e-4}$ |
> |--------------|-------------|-------------|-------------|-------------|-------------|-------------|
> | SFT          | 67.8        | 76.1        | 77.9        | 94.61       | 94.46       | 94.27       |
> | DeepAlign    | 25.9        | 58.2        | 71.4        | 93.12       | 93.35       | 93.12       |
> | LISA         | 52.5        | 69.0        | 78.5        | 94.27       | 94.15       | 93.81       |
> | SBR ($\lambda$=50)    | 5.8    | 47.5        | 62.5        | 94.15       | 94.27       | 94.15       |
> | SBR ($\lambda$=100)  | 6.4         | 8.6         | 22.7        | 93.92       | 93.35       | 94.38       |
> | SBR ($\lambda$=200)  | 5.1         | 6.2         | 7.1         | 93.81       | 93.46       | 93.69       |
>
> As shown, when the learning rate reaches $10^{-4}$, all methods suffer safety collapse. In contrast, SBR maintains robust defense under aggressive learning rates by increasing the regularization strength $\lambda$. This result aligns with our conclusion in Section 4.3 [L323, C1]: increasing $\lambda$ imposes a stronger defense, realizing a controllable safety-utility trade-off.
>
> ---
>
> ### **Comment 2: Evaluate on newer models (Qwen3, Gemma3)**
> SBR targets the unembedding layer, a fundamental architectural feature shared across Transformer-based LLMs. Consequently, scaling parameters or updating versions do not alter its core mechanism.
>
> To address the reviewer's suggestion, we prioritize scaling up to Qwen3-14B, Gemma3-13B. Given the length limitation, we focused on larger models as they typically possess stronger instruction-following capabilities, presenting a more realistic testbed for defense mechanisms than sub-7B models. (poison ratios $p$ = 0.1 and 0.3):
>
> **Table 2: Gemma3-13B**
> | Method       | HS$_{p=0.1}$ | HS$_{p=0.3}$ | FA$_{p=0.1}$ | FA$_{p=0.3}$ |
> |--------------|---------------|---------------|---------------|---------------|
> | SFT          | 77.8          | 80.4          | 94.84         | 93.35         |
> | DeepAlign    | 19.5          | 35.2          | 92.43         | 91.18         |
> | LISA         | 58.4          | 71.3          | 93.12         | 93.81         |
> | SBR (Ours)   | 8.0           | 12.3          | 93.58         | 93.69         |
>
> **Table 3: Qwen3-14B**
> | Method       | HS$_{p=0.1}$ | HS$_{p=0.3}$ | FA$_{p=0.1}$ | FA$_{p=0.3}$ |
> |--------------|---------------|---------------|---------------|---------------|
> | SFT          | 74.2          | 76.0          | 95.41         | 95.18         |
> | DeepAlign    | 19.1          | 32.3          | 94.72         | 93.69         |
> | LISA         | 67.7          | 66.0          | 92.59         | 92.66         |
> | SBR (Ours)   | 6.4           | 10.1          | 94.61         | 94.72         |
>
> In summary, SBR maintains HS<13 across all attack intensities on both advanced models. This supplementary evaluation yields no contradictory findings. Instead, it strictly aligns with our original conclusions on [L309, C1]: the geometric bottleneck ensures SBR remains robust across diverse model scales and architectures.
>
> ---
>
> ### **Comment 3: Evaluate against GCG**
> We thank the reviewer for this thoughtful question about the scope of our method. We clarify:
> 1. Our core threat model is **Harmful Fine-Tuning (HFT)** (malicious parameter updates) [1-3]. GCG is an input-space adversarial attack that optimizes prompts **without modifying model parameters**, a fundamentally distinct paradigm.
> 2. SBR is designed to anchor safety during parameter updates, with no native capability against input-space attacks, consistent with all prior HFT defense work [1-3].
>
> We will add this scope clarification to the Limitations section, and note that combining SBR with dedicated input-space defenses is a promising future direction.
>
> ---
>
> [1] LISA: Lazy safety alignment for LLM against HFT. NeurIPS 2024
>
> [2] Vaccine: Perturbation-aware alignment for LLM against HFT. NeurIPS 2024
>
> [3] Fine-tuning aligned language models compromises safety, even when users do not intend to! ICLR 2024
>
> We hope these address all your concerns. Further details are available upon request.

---

> > ### Author Rebuttal · Reviewer_CUkh · 2026-04-03
> >
> > My questions are fully resolved. Thank you to the authors for your efforts.

---

> > > ### Author Response · Authors · 2026-04-04
> > >
> > > We appreciate your continued assessment and are glad that your main concerns have been addressed.

---

### Decision · Program_Chairs · 2026-04-30

**Decision:**

Accept (regular)

**Comment:**

This paper identifies parameter redundancy as the main cause of defense failure. They propose a new perspective that shifts defense from the entire large-scale parameter space to the more restricted output bottleneck. Instead of trying to control how the model learns, internal to its weights, the paper proposes a safety-bottleneck regularization method to anchor the model’s final hidden state and control what the model must output. Experiments show that existing defenses collapse quickly under persistent fine-tuning, while SBR can keep a harmful score < 10 on a variety of tasks.

The reviewers proposed several good reasons for accepting the paper:
- Simple and efficient method that is easy to deploy in practice.
- Strong empirical results and ablation studies.
- Good and clear theoretical motivations.

The reviewers also offer several constructive comments to improve the paper, including limited model coverage, limited harmful datasets, outdated models, limited poisoning ratios, LoRA fine-tuning, and a need for more attack evaluations. Many of these issues have been successfully addressed during the rebuttal.